# HYPE: A Benchmark for Human eYe Perceptual Evaluation of Generative Models

**Sharon Zhou**[*]**, Mitchell L. Gordon**[*]**, Ranjay Krishna,**
**Austin Narcomey, Li Fei-Fei, Michael S. Bernstein**
Stanford University
{sharonz, mgord, ranjaykrishna, aon2, feifeili, msb}@cs.stanford.edu

## Abstract

Generative models often use human evaluations to measure the perceived quality of their outputs. Automated metrics are noisy indirect proxies, because they rely on heuristics or pretrained embeddings. However, up until now, direct human evaluation strategies have been ad-hoc, neither standardized nor validated. Our work establishes a gold standard human benchmark for generative realism. We construct HUMAN EYE PERCEPTUAL EVALUATION (HYPE), a human benchmark that is (1) *grounded* in psychophysics research in perception, (2) *reliable* across different sets of randomly sampled outputs from a model, (3) able to produce *separable* model performances, and (4) *efficient* in cost and time. We introduce two variants: one that measures visual perception under adaptive time constraints to determine the threshold at which a model's outputs appear real (e.g. 250ms), and the other a less expensive variant that measures human error rate on fake and real images sans time constraints. We test HYPE across six state-of-the-art generative adversarial networks and two sampling techniques on conditional and unconditional image generation using four datasets: CelebA, FFHQ, CIFAR-10, and ImageNet. We find that HYPE can track the relative improvements between models, and we confirm via bootstrap sampling that these measurements are consistent and replicable.

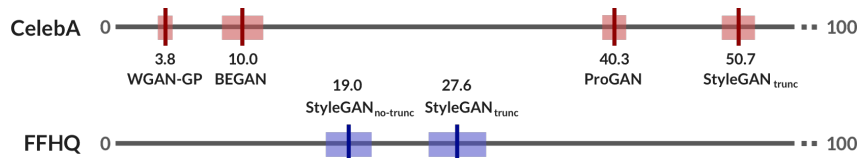

Figure 1: Our human evaluation metric, HYPE, consistently distinguishes models from each other: here, we compare different generative models performance on FFHQ. A score of 50% represents indistinguishable results from real, while a score above 50% represents hyper-realism.

## 1 Introduction

Generating realistic images is regarded as a focal task for measuring the progress of generative models. Automated metrics are either heuristic approximations [49, 52, 14, 26, 9, 45] or intractable density estimations, examined to be inaccurate on high dimensional problems [24, 7, 55]. Human evaluations, such as those given on Amazon Mechanical Turk [49, 14], remain ad-hoc because "results change drastically" [52] based on details of the task design [36, 34, 27]. With both noisy automated and noisy human benchmarks, measuring progress over time has become akin to hill-climbing on noise. Even widely used metrics, such as Inception Score [52] and Fréchet Inception Distance [23], have been discredited for their application to non-ImageNet datasets [3, 48, 8, 46]. Thus, to monitor progress,

---

[*]Equal contribution.

generative models need a systematic gold standard benchmark. In this paper, we introduce a gold standard benchmark for realistic generation, demonstrating its effectiveness across four datasets, six models, and two sampling techniques, and using it to assess the progress of generative models over time.

Realizing the constraints of available automated metrics, many generative modeling tasks resort to human evaluation and visual inspection [49, 52, 14]. These human measures are (1) ad-hoc, each executed in idiosyncrasy without proof of reliability or grounding to theory, and (2) high variance in their estimates [52, 14, 42]. These characteristics combine to a lack of reliability, and downstream, (3) a lack of clear separability between models. Theoretically, given sufficiently large sample sizes of human evaluators and model outputs, the law of large numbers would smooth out the variance and reach eventual convergence; but this would occur at (4) a high cost and a long delay.

We present HYPE (HUMAN EYE PERCEPTUAL EVALUATION) to address these criteria in turn. HYPE: (1) measures the perceptual realism of generative model outputs via a **grounded** method inspired by psychophysics methods in perceptual psychology, (2) is a **reliable** and consistent estimator, (3) is statistically **separable** to enable a comparative ranking, and (4) ensures a cost and time **efficient** method through modern crowdsourcing techniques such as training and aggregation. We present two methods of evaluation. The first, called $HYPE_{time}$, is inspired directly by the psychophysics literature [28, 11], and displays images using adaptive time constraints to determine the time-limited perceptual threshold a person needs to distinguish real from fake. The $HYPE_{time}$ score is understood as the minimum time, in milliseconds, that a person needs to see the model's output before they can distinguish it as real or fake. For example, a score of $500$ms on $HYPE_{time}$ indicates that humans can distinguish model outputs from real images at $500$ms exposure times or longer, but not under $500$ms. The second method, called $HYPE_\infty$, is derived from the first to make it simpler, faster, and cheaper while maintaining reliability. It is interpretable as the rate at which people mistake fake images and real images, given unlimited time to make their decisions. A score of $50\%$ on $HYPE_\infty$ means that people differentiate generated results from real data at chance rate, while a score above $50\%$ represents hyper-realism in which generated images appear more real than real images.

We run two large-scale experiments. First, we demonstrate HYPE's performance on unconditional human face generation using four popular generative adversarial networks (GANs) [20, 5, 25, 26] across CelebA-64 [37]. We also evaluate two newer GANs [41, 9] on FFHQ-1024 [26]. HYPE indicates that GANs have clear, measurable perceptual differences between them; this ranking is identical in both $HYPE_{time}$ and $HYPE_\infty$. The best performing model, StyleGAN trained on FFHQ and sampled with the truncation trick, only performs at $27.6\%$ $HYPE_\infty$, suggesting substantial opportunity for improvement. We can reliably reproduce these results with $95\%$ confidence intervals using 30 human evaluators at $\$60$ in a task that takes 10 minutes.

Second, we demonstrate the performance of $HYPE_\infty$ beyond faces on conditional generation of five object classes in ImageNet [13] and unconditional generation of CIFAR-10 [31]. Early GANs such as BEGAN are not separable in $HYPE_\infty$ when generating CIFAR-10: none of them produce convincing results to humans, verifying that this is a harder task than face generation. The newer StyleGAN shows separable improvement, indicating progress over the previous models. With ImageNet-5, GANs have improved on classes considered "easier" to generate (e.g., lemons), but resulted in consistently low scores across all models for harder classes (e.g., French horns).

HYPE is a rapid solution for researchers to measure their generative models, requiring just a single click to produce reliable scores and measure progress. We deploy HYPE at https://hype.stanford.edu, where researchers can upload a model and retrieve a HYPE score. Future work will extend HYPE to additional generative tasks such as text, music, and video generation.

## 2   HYPE: A benchmark for HUMAN EYE PERCEPTUAL EVALUATION

HYPE displays a series of images one by one to crowdsourced evaluators on Amazon Mechanical Turk and asks the evaluators to assess whether each image is real or fake. Half of the images are real images, drawn from the model's training set (e.g., FFHQ, CelebA, ImageNet, or CIFAR-10). The other half are drawn from the model's output. We use modern crowdsourcing training and quality control techniques [40] to ensure high-quality labels. Model creators can choose to perform two different evaluations: $HYPE_{time}$, which gathers time-limited perceptual thresholds to measure the

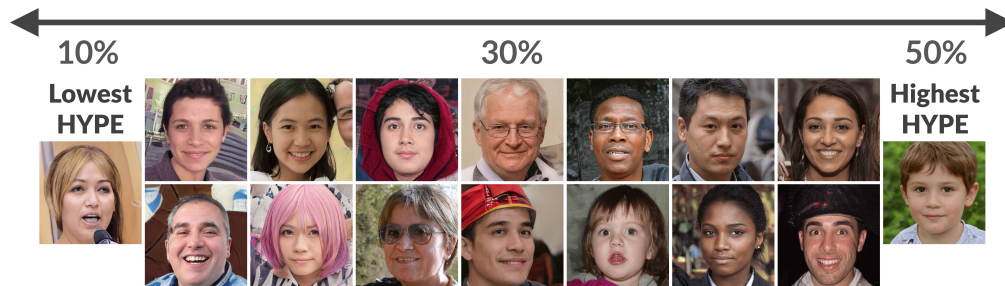

**10%**                              **30%**                              **50%**

**Lowest HYPE**                                                    **Highest HYPE**

Figure 2: Example images sampled with the truncation trick from StyleGAN trained on FFHQ. Images on the right exhibit the highest $HYPE_\infty$ scores, the highest human perceptual fidelity.

psychometric function and report the minimum time people need to make accurate classifications, and $HYPE_\infty$, a simplified approach which assesses people's error rate under no time constraint.

## 2.1 $HYPE_{time}$: Perceptual fidelity grounded in psychophysics

Our first method, $HYPE_{time}$, measures time-limited perceptual thresholds. It is rooted in psychophysics literature, a field devoted to the study of how humans perceive stimuli, to evaluate human time thresholds upon perceiving an image. Our evaluation protocol follows the procedure known as the *adaptive staircase method* (Figure 3) [11]. An image is flashed for a limited length of time, after which the evaluator is asked to judge whether it is real or fake. If the evaluator consistently answers correctly, the staircase descends and flashes the next image with less time. If the evaluator is incorrect, the staircase ascends and provides more time.

This process requires sufficient iterations to converge to the evaluator's perceptual threshold: the shortest exposure time at which they can maintain effective performance [11, 19, 15]. The process produces what is known as the *psychometric function* [60], the relationship of timed stimulus exposure to accuracy. For example, for an easily distinguishable set of generated images, a human evaluator would immediately drop to the lowest millisecond exposure.

$HYPE_{time}$ displays three blocks of staircases for each evaluator. An image evaluation begins with a 3-2-1 countdown clock, each number displaying for 500ms [30]. The sampled image is then displayed for the current exposure time. Immediately after each image, four perceptual mask images are rapidly displayed for 30ms each. These noise masks are distorted to prevent retinal afterimages and further sensory processing after the image disappears [19]. We generate masks using an existing texture-synthesis algorithm [44]. Upon each submission, $HYPE_{time}$ reveals to the evaluator whether they were correct.

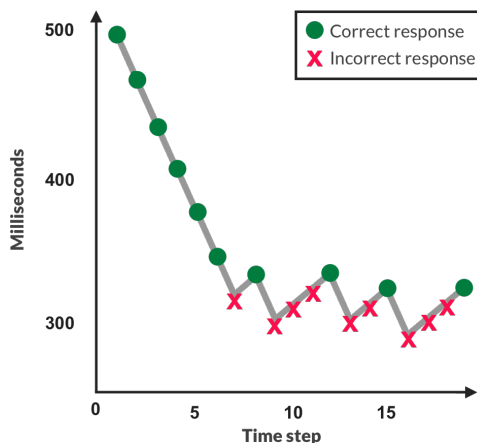

Figure 3: The adaptive staircase method shows images to evaluators at different time exposures, decreasing when correct and increasing when incorrect. The modal exposure measures their perceptual threshold.

Image exposures are in the range [100ms, 1000ms], derived from the perception literature [17]. All blocks begin at 500ms and last for 150 images (50% generated, 50% real), values empirically tuned from prior work [11, 12]. Exposure times are raised at 10ms increments and reduced at 30ms decrements, following the 3-up/1-down adaptive staircase approach, which theoretically leads to a 75% accuracy threshold that approximates the human perceptual threshold [35, 19, 11].

Every evaluator completes multiple staircases, called *blocks*, on different sets of images. As a result, we observe multiple measures for the model. We employ three blocks, to balance quality estimates against evaluators' fatigue [32, 50, 22]. We average the modal exposure times across blocks to

calculate a final value for each evaluator. Higher scores indicate a better model, whose outputs take longer time exposures to discern from real.

## 2.2 HYPE$_\infty$: Cost-effective approximation

Building on the previous method, we introduce HYPE$_\infty$: a simpler, faster, and cheaper method after ablating HYPE$_{time}$ to optimize for speed, cost, and ease of interpretation. HYPE$_\infty$ shifts from a measure of perceptual time to a measure of human deception rate, given infinite evaluation time. The HYPE$_\infty$ score gauges total error on a task of 50 fake and 50 real images [2], enabling the measure to capture errors on both fake and real images, and effects of hyperrealistic generation when fake images look even more realistic than real images [3]. HYPE$_\infty$ requires fewer images than HYPE$_{time}$ to find a stable value, empirically producing a 6x reduction in time and cost (10 minutes per evaluator instead of 60 minutes, at the same rate of \$12 per hour). Higher scores are again better: 10% HYPE$_\infty$ indicates that only 10% of images deceive people, whereas 50% indicates that people are mistaking real and fake images at chance, rendering fake images indistinguishable from real. Scores above 50% suggest hyperrealistic images, as evaluators mistake images at a rate greater than chance.

HYPE$_\infty$ shows each evaluator a total of 100 images: 50 real and 50 fake. We calculate the proportion of images that were judged incorrectly, and aggregate the judgments over the $n$ evaluators on $k$ images to produce the final score for a given model.

## 2.3 Consistent and reliable design

To ensure that our reported scores are consistent and reliable, we need to sample sufficiently from the model as well as hire, qualify, and appropriately pay enough evaluators.

**Sampling sufficient model outputs.** The selection of $K$ images to evaluate from a particular model is a critical component of a fair and useful evaluation. We must sample a large enough number of images that fully capture a model's generative diversity, yet balance that against tractable costs in the evaluation. We follow existing work on evaluating generative output by sampling $K = 5000$ generated images from each model [52, 41, 58] and $K = 5000$ real images from the training set. From these samples, we randomly select images to give to each evaluator.

**Quality of evaluators.** To obtain a high-quality pool of evaluators, each is required to pass a qualification task. Such a pre-task filtering approach, sometimes referred to as a person-oriented strategy, is known to outperform process-oriented strategies that perform post-task data filtering or processing [40]. Our qualification task displays 100 images (50 real and 50 fake) with no time limits. Evaluators must correctly classify 65% of both real and fake images. This threshold should be treated as a hyperparameter and may change depending upon the GANs used in the tutorial and the desired discernment ability of the chosen evaluators. We choose 65% based on the cumulative binomial probability of 65 binary choice answers out of 100 total answers: there is only a one in one-thousand chance that an evaluator will qualify by random guessing. Unlike in the task itself, fake qualification images are drawn equally from multiple different GANs to ensure an equitable qualification across all GANs. The qualification is designed to be taken occasionally, such that a pool of evaluators can assess new models on demand.

**Payment.** Evaluators are paid a base rate of \$1 for working on the qualification task. To incentivize evaluators to remained engaged throughout the task, all further pay after the qualification comes from a bonus of \$0.02 per correctly labeled image, typically totaling a wage of \$12/hr.

## 3 Experimental setup

**Datasets.** We evaluate on four datasets. (1) CelebA-64 [37] is popular dataset for unconditional image generation with 202k images of human faces, which we align and crop to be $64 \times 64$ px. (2) FFHQ-1024 [26] is a newer face dataset with 70k images of size $1024 \times 1024$ px. (3) CIFAR-10

consists of 60k images, sized $32 \times 32$ px, across 10 classes. (4) ImageNet-5 is a subset of 5 classes with 6.5k images at $128 \times 128$ px from the ImageNet dataset [13], which have been previously identified as easy (lemon, Samoyed, library) and hard (baseball player, French horn) [9].

**Architectures.** We evaluate on four state-of-the-art models trained on CelebA-64 and CIFAR-10: StyleGAN [26], ProGAN [25], BEGAN [5], and WGAN-GP [20]. We also evaluate on two models, SN-GAN [41] and BigGAN [9] trained on ImageNet, sampling conditionally on each class in ImageNet-5. We sample BigGAN with ($\sigma = 0.5$ [9]) and without the truncation trick.

We also evaluate on StyleGAN [26] trained on FFHQ-1024 with ($\psi = 0.7$ [26]) and without truncation trick sampling. For parity on our best models across datasets, StyleGAN instances trained on CelebA-64 and CIFAR-10 are also sampled with the truncation trick.

We sample noise vectors from the $d$-dimensional spherical Gaussian noise prior $z \in \mathbb{R}^d \sim \mathcal{N}(0, I)$ during training and test times. We specifically opted to use the same standard noise prior for comparison, yet are aware of other priors that optimize for FID and IS scores [9]. We select training hyperparameters published in the corresponding papers for each model.

**Evaluator recruitment.** We recruit 930 evaluators from Amazon Mechanical Turk, or 30 for each run of HYPE. We explain our justification for this number in the Cost tradeoffs section. To maintain a between-subjects study in this evaluation, we recruit independent evaluators across tasks and methods.

**Metrics.** For $\text{HYPE}_{\text{time}}$, we report the modal perceptual threshold in milliseconds. For $\text{HYPE}_\infty$, we report the error rate as a percentage of images, as well as the breakdown of this rate on real and fake images separately. To show that our results for each model are separable, we report a one-way ANOVA with Tukey pairwise post-hoc tests to compare all models.

Reliability is a critical component of HYPE, as a benchmark is not useful if a researcher receives a different score when rerunning it. We use bootstrapping [16], repeated resampling from the empirical label distribution, to measure variation in scores across multiple samples with replacement from a set of labels. We report $95\%$ bootstrapped confidence intervals (CIs), along with standard deviation of the bootstrap sample distribution, by randomly sampling 30 evaluators with replacement from the original set of evaluators across $10,000$ iterations.

**Experiment 1:** We run two large-scale experiments to validate HYPE. The first one focuses on the controlled evaluation and comparison of $\text{HYPE}_{\text{time}}$ against $\text{HYPE}_\infty$ on established human face datasets. We recorded responses totaling (4 CelebA-64 + 2 FFHQ-1024) models $\times$ 30 evaluators $\times$ 550 responses = 99k total responses for our $\text{HYPE}_{\text{time}}$ evaluation and (4

Table 1: $\text{HYPE}_{\text{time}}$ on $\text{StyleGAN}_{\text{trunc}}$ and $\text{StyleGAN}_{\text{no-trunc}}$ trained on FFHQ-1024.

| Rank | GAN | $\text{HYPE}_{\text{time}}$ (ms) | Std. | 95% CI |
|------|-----|------------------------|------|--------|
| 1 | $\text{StyleGAN}_{\text{trunc}}$ | 363.2 | 32.1 | $300.0 - 424.3$ |
| 2 | $\text{StyleGAN}_{\text{no-trunc}}$ | 240.7 | 29.9 | $184.7 - 302.7$ |

CelebA-64 + 2 FFHQ-1024) models $\times$ 30 evaluators $\times$ 100 responses = 18k, for our $\text{HYPE}_\infty$ evaluation.

**Experiment 2:** The second experiment evaluates $\text{HYPE}_\infty$ on general image datasets. We recorded (4 CIFAR-10 + 3 ImageNet-5) models $\times$ 30 evaluators $\times$ 100 responses = 57k total responses.

## 4   Experiment 1: $\text{HYPE}_{\text{time}}$ and $\text{HYPE}_\infty$ on human faces

We report results on $\text{HYPE}_{\text{time}}$ and demonstrate that the results of $\text{HYPE}_\infty$ approximates those from $\text{HYPE}_{\text{time}}$ at a fraction of the cost and time.

### 4.1   $\text{HYPE}_{\text{time}}$

**CelebA-64.** We find that $\text{StyleGAN}_{\text{trunc}}$ resulted in the highest $\text{HYPE}_{\text{time}}$ score (modal exposure time), at a mean of $439.3$ms, indicating that evaluators required nearly a half-second of exposure to accurately classify $\text{StyleGAN}_{\text{trunc}}$ images (Table 1). $\text{StyleGAN}_{\text{trunc}}$ is followed by ProGAN at $363.7$ms, a $17\%$ drop in time. BEGAN and WGAN-GP are both easily identifiable as fake, tied in last place around the minimum available exposure time of 100ms. Both BEGAN and WGAN-GP exhibit a bottoming out effect — reaching the minimum time exposure of 100ms quickly and consistently.[4]

To demonstrate separability between models we report results from a one-way analysis of variance (ANOVA) test, where each model's input is the list of modes from each model's 30 evaluators. The ANOVA results confirm that there is a statistically significant omnibus difference ($F(3, 29) = 83.5, p < 0.0001$). Pairwise post-hoc analysis using Tukey tests confirms that all pairs of models are separable (all $p < 0.05$) except BEGAN and WGAN-GP ($n.s.$).

**FFHQ-1024**. We find that $\text{StyleGAN}_{\text{trunc}}$ resulted in a higher exposure time than $\text{StyleGAN}_{\text{no-trunc}}$, at 363.2ms and 240.7ms, respectively (Table 1). While the $95\%$ confidence intervals that represent a very conservative overlap of 2.7ms, an unpaired t-test confirms that the difference between the two models is significant ($t(58) = 2.3, p = 0.02$).

## 4.2 HYPE$_\infty$

**CelebA-64**. Table 2 reports results for HYPE$_\infty$ on CelebA-64. We find that $\text{StyleGAN}_{\text{trunc}}$ resulted in the highest HYPE$_\infty$ score, fooling evaluators $50.7\%$ of the time. $\text{StyleGAN}_{\text{trunc}}$ is followed by ProGAN at $40.3\%$, BEGAN at $10.0\%$, and WGAN-GP at $3.8\%$. No confidence intervals are overlapping and an ANOVA test is significant ($F(3, 29) = 404.4, p < 0.001$). Pairwise post-hoc Tukey tests show that all pairs of models are separable (all $p < 0.05$). Notably, HYPE$_\infty$ results in separable results for BEGAN and WGAN-GP, unlike in HYPE$_{\text{time}}$ where they were not separable due to a bottoming-out effect.

Table 2: HYPE$_\infty$ on four GANs trained on CelebA-64. Counterintuitively, real errors increase with the errors on fake images, because evaluators become more confused and distinguishing factors between the two distributions become harder to discern.

| Rank | GAN | HYPE$_\infty$ (%) | Fakes Error | Reals Error | Std. | 95% CI | KID | FID | Precision |
|---|---|---|---|---|---|---|---|---|---|
| 1 | StyleGAN$_{\text{trunc}}$ | 50.7% | 62.2% | 39.3% | 1.3 | $48.2 - 53.1$ | 0.005 | 131.7 | 0.982 |
| 2 | ProGAN | 40.3% | 46.2% | 34.4% | 0.9 | $38.5 - 42.0$ | 0.001 | 2.5 | 0.990 |
| 3 | BEGAN | 10.0% | 6.2% | 13.8% | 1.6 | $7.2 - 13.3$ | 0.056 | 67.7 | 0.326 |
| 4 | WGAN-GP | 3.8% | 1.7% | 5.9% | 0.6 | $3.2 - 5.7$ | 0.046 | 43.6 | 0.654 |

**FFHQ-1024**. We observe a consistently separable difference between $\text{StyleGAN}_{\text{trunc}}$ and $\text{StyleGAN}_{\text{no-trunc}}$ and clear delineations between models (Table 3). HYPE$_\infty$ ranks $\text{StyleGAN}_{\text{trunc}}$ ($27.6\%$) above $\text{StyleGAN}_{\text{no-trunc}}$ ($19.0\%$) with no overlapping CIs. Separability is confirmed by an unpaired t-test ($t(58) = 8.3, p < 0.001$).

Table 3: HYPE$_\infty$ on $\text{StyleGAN}_{\text{trunc}}$ and $\text{StyleGAN}_{\text{no-trunc}}$ trained on FFHQ-1024. Evaluators were deceived most often by $\text{StyleGAN}_{\text{trunc}}$. Similar to CelebA-64, fake errors and real errors track each other as the line between real and fake distributions blurs.

| Rank | GAN | HYPE$_\infty$ (%) | Fakes Error | Reals Error | Std. | 95% CI | KID | FID | Precision |
|---|---|---|---|---|---|---|---|---|---|
| 1 | StyleGAN$_{\text{trunc}}$ | 27.6% | 28.4% | 26.8% | 2.4 | $22.9 - 32.4$ | 0.007 | 13.8 | 0.976 |
| 2 | StyleGAN$_{\text{no-trunc}}$ | 19.0% | 18.5% | 19.5% | 1.8 | $15.5 - 22.4$ | 0.001 | 4.4 | 0.983 |

## 4.3 Cost tradeoffs with accuracy and time

One of HYPE's goals is to be cost and time efficient. When running HYPE, there is an inherent tradeoff between accuracy and time, as well as between accuracy and cost. This is driven by the law of large numbers: recruiting additional evaluators in a crowdsourcing task often produces more consistent results, but at a higher cost (as each evaluator is paid for their work) and a longer amount of time until completion (as more evaluators must be recruited and they must complete their work).

To manage this tradeoff, we run an experiment with HYPE$_\infty$ on $\text{StyleGAN}_{\text{trunc}}$. We completed an additional evaluation with 60 evaluators, and compute $95\%$ bootstrapped confidence intervals, choosing from 10 to 120 evaluators (Figure 4). We see that the CI begins to converge around 30 evaluators, our recommended number of evaluators to recruit.

Payment to evaluators was calculated as described in the Approach section. At 30 evaluators, the cost of running HYPE$_{\text{time}}$ on one model was approximately \$360, while the cost of running HYPE$_\infty$ on the same model was approximately \$60. Payment per evaluator for both tasks was approximately

$12/hr. Evaluators spent an average of one hour each on a HYPE$_{time}$ task and 10 minutes each on a HYPE$_\infty$ task. Thus, HYPE$_\infty$ achieves its goals of being significantly cheaper to run, while maintaining consistency.

## 4.4 Comparison to automated metrics

As FID [23] is one of the most frequently used evaluation methods for unconditional image generation, it is imperative to compare HYPE against FID on the same models. We also compare to two newer automated metrics: KID [6], an unbiased estimator independent of sample size, and $F_{1/8}$ (precision) [51], which captures fidelity independently. We show through Spearman rank-order correlation coefficients that HYPE scores are not correlated with FID ($\rho = -0.029, p = 0.96$), where a Spearman correlation of $-1.0$ is ideal because lower FID and higher HYPE scores indicate stronger models. We therefore find that FID is not highly correlated with human judgment. Meanwhile, HYPE$_{time}$ and HYPE$_\infty$ exhibit strong correlation ($\rho = 1.0, p = 0.0$), where $1.0$ is ideal because they are directly related. We calculate FID across the standard protocol of 50K generated and 50K real images for both CelebA-64 and FFHQ-1024, reproducing scores for StyleGAN$_{no\text{-}trunc}$. KID ($\rho = -0.609, p = 0.20$) and precision ($\rho = 0.657, p = 0.16$) both show a statistically insignificant but medium level of correlation with humans.

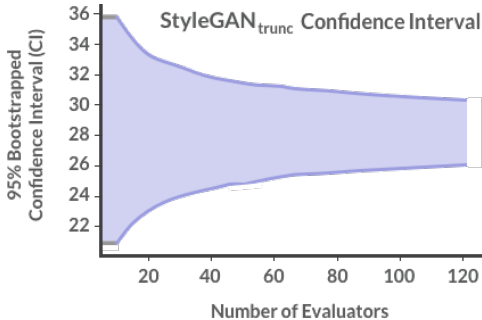

Figure 4: Effect of more evaluators on CI.

## 4.5 HYPE$_\infty$ during model training

HYPE can also be used to evaluate progress during model training. We find that HYPE$_\infty$ scores increased as StyleGAN training progressed from $29.5\%$ at 4k epochs, to $45.9\%$ at 9k epochs, to $50.3\%$ at 25k epochs ($F(2, 29) = 63.3, p < 0.001$).

# 5 Experiment 2: HYPE$_\infty$ beyond faces

We now turn to another popular image generation task: objects. As Experiment 1 showed HYPE$_\infty$ to be an efficient and cost effective variant of HYPE$_{time}$, here we focus exclusively on HYPE$_\infty$.

## 5.1 ImageNet-5

We evaluate conditional image generation on five ImageNet classes (Table 4). We also report FID [23], KID [6], and $F_{1/8}$ (precision) [51] scores. To evaluate the relative effectiveness of the three GANs within each object class, we compute five one-way ANOVAs, one for each of the object classes. We find that the HYPE$_\infty$ scores are separable for images from three easy classes: samoyeds (dogs) ($F(2, 29) = 15.0, p < 0.001$), lemons ($F(2, 29) = 4.2, p = 0.017$), and libraries ($F(2, 29) = 4.9, p = 0.009$). Pairwise Posthoc tests reveal that this difference is only significant between SN-GAN and the two BigGAN variants. We also observe that models have unequal strengths, e.g. SN-GAN is better suited to generating libraries than samoyeds.

**Comparison to automated metrics**. Spearman rank-order correlation coefficients on all three GANs across all five classes show that there is a low to moderate correlation between the HYPE$_\infty$ scores and KID ($\rho = -0.377, p = 0.02$), FID ($\rho = -0.282, p = 0.01$), and negligible correlation with precision ($\rho = -0.067, p = 0.81$). Some correlation for our ImageNet-5 task is expected, as these metrics use pretrained ImageNet embeddings to measure differences between generated and real data.

Interestingly, we find that this correlation depends upon the GAN: considering only SN-GAN, we find stronger coefficients for KID ($\rho = -0.500, p = 0.39$), FID ($\rho = -0.300, p = 0.62$), and precision ($\rho = -0.205, p = 0.74$). When considering only BigGAN, we find far weaker coefficients for KID ($\rho = -0.151, p = 0.68$), FID ($\rho = -0.067, p = .85$), and precision ($\rho = -0.164, p = 0.65$). This

illustrates an important flaw with these automatic metrics: their ability to correlate with humans depends upon the generative model that the metrics are evaluating on, varying by model and by task.

Table 4: HYPE$_\infty$ on three models trained on ImageNet and conditionally sampled on five classes. BigGAN routinely outperforms SN-GAN. BigGan$_{trunc}$ and BigGan$_{no-trunc}$ are not separable.

| | GAN | Class | HYPE$_\infty$ (%) | Fakes Error | Reals Error | Std. | 95% CI | KID | FID | Precision |
|---|---|---|---|---|---|---|---|---|---|---|
| Easy | BigGan$_{trunc}$ | Lemon | 18.4% | 21.9% | 14.9% | 2.3 | 14.2–23.1 | 0.043 | 94.22 | 0.784 |
| | BigGan$_{no-trunc}$ | Lemon | 20.2% | 22.2% | 18.1% | 2.2 | 16.0–24.8 | 0.036 | 87.54 | 0.774 |
| | SN-GAN | Lemon | 12.0% | 10.8% | 13.3% | 1.6 | 9.0–15.3 | 0.053 | 117.90 | 0.656 |
| Easy | BigGan$_{trunc}$ | Samoyed | 19.9% | 23.5% | 16.2% | 2.6 | 15.0–25.1 | 0.027 | 56.94 | 0.794 |
| | BigGan$_{no-trunc}$ | Samoyed | 19.7% | 23.2% | 16.1% | 2.2 | 15.5–24.1 | 0.014 | 46.14 | 0.906 |
| | SN-GAN | Samoyed | 5.8% | 3.4% | 8.2% | 0.9 | 4.1–7.8 | 0.046 | 88.68 | 0.785 |
| Easy | BigGan$_{trunc}$ | Library | 17.4% | 22.0% | 12.8% | 2.1 | 13.3–21.6 | 0.049 | 98.45 | 0.695 |
| | BigGan$_{no-trunc}$ | Library | 22.9% | 28.1% | 17.6% | 2.1 | 18.9–27.2 | 0.029 | 78.49 | 0.814 |
| | SN-GAN | Library | 13.6% | 15.1% | 12.1% | 1.9 | 10.0–17.5 | 0.043 | 94.89 | 0.814 |
| Hard | BigGan$_{trunc}$ | French Horn | 7.3% | 9.0% | 5.5% | 1.8 | 4.0–11.2 | 0.031 | 78.21 | 0.732 |
| | BigGan$_{no-trunc}$ | French Horn | 6.9% | 8.6% | 5.2% | 1.4 | 4.3–9.9 | 0.042 | 96.18 | 0.757 |
| | SN-GAN | French Horn | 3.6% | 5.0% | 2.2% | 1.0 | 1.8–5.9 | 0.156 | 196.12 | 0.674 |
| Hard | BigGan$_{trunc}$ | Baseball Player | 1.9% | 1.9% | 1.9% | 0.7 | 0.8–3.5 | 0.049 | 91.31 | 0.853 |
| | BigGan$_{no-trunc}$ | Baseball Player | 2.2% | 3.3% | 1.2% | 0.6 | 1.3–3.5 | 0.026 | 76.71 | 0.838 |
| | SN-GAN | Baseball Player | 2.8% | 3.6% | 1.9% | 1.5 | 0.8–6.2 | 0.052 | 105.82 | 0.785 |

Table 5: Four models on CIFAR-10. StyleGAN$_{trunc}$ can generate realistic images from CIFAR-10.

| GAN | HYPE$_\infty$ (%) | Fakes Error | Reals Error | Std. | 95% CI | KID | FID | Precision |
|---|---|---|---|---|---|---|---|---|
| StyleGAN$_{trunc}$ | 23.3% | 28.2% | 18.5% | 1.6 | 20.1–26.4 | 0.005 | 62.9 | 0.982 |
| PROGAN | 14.8% | 18.5% | 11.0% | 1.6 | 11.9–18.0 | 0.001 | 53.2 | 0.990 |
| BEGAN | 14.5% | 14.6% | 14.5% | 1.7 | 11.3–18.1 | 0.056 | 96.2 | 0.326 |
| WGAN-GP | 13.2% | 15.3% | 11.1% | 2.3 | 9.1–18.1 | 0.046 | 104.0 | 0.654 |

## 5.2 CIFAR-10

For the difficult task of unconditional generation on CIFAR-10, we use the same four model architectures in Experiment 1: CelebA-64. Table 5 shows that HYPE$_\infty$ was able to separate StyleGAN$_{trunc}$ from the earlier BEGAN, WGAN-GP, and ProGAN, indicating that StyleGAN is the first among them to make human-perceptible progress on unconditional object generation with CIFAR-10.

**Comparison to automated metrics**. Spearman rank-order correlation coefficients on all four GANs show medium, yet statistically insignificant, correlations with KID ($\rho = -0.600, p = 0.40$) and FID ($\rho = 0.600, p = 0.40$) and precision ($\rho = -.800, p = 0.20$).

## 6 Related work

**Cognitive psychology.** We leverage decades of cognitive psychology to motivate how we use stimulus timing to gauge the perceptual realism of generated images. It takes an average of 150ms of focused visual attention for people to process and interpret an image, but only 120ms to respond to faces because our inferotemporal cortex has dedicated neural resources for face detection [47, 10]. Perceptual masks are placed between a person's response to a stimulus and their perception of it to eliminate post-processing of the stimuli after the desired time exposure [53]. Prior work in determining human perceptual thresholds [19] generates masks from their test images using the texture-synthesis algorithm [44]. We leverage this literature to establish feasible lower bounds on the exposure time of images, the time between images, and the use of noise masks.

**Success of automatic metrics.** Common generative modeling tasks include realistic image generation [18], machine translation [1], image captioning [57], and abstract summarization [39], among others. These tasks often resort to automatic metrics like the Inception Score (IS) [52] and Fréchet Inception Distance (FID) [23] to evaluate images and BLEU [43], CIDEr [56] and METEOR [2] scores to evaluate text. While we focus on how realistic generated content appears, other automatic metrics also measure diversity of output, overfitting, entanglement, training stability, and computational and sample efficiency of the model [8, 38, 3]. Our metric may also capture one aspect of output diversity,

insofar as human evaluators can detect similarities or patterns across images. Our evaluation is not meant to replace existing methods but to complement them.

**Limitations of automatic metrics.** Prior work has asserted that there exists coarse correlation of human judgment to FID [23] and IS [52], leading to their widespread adoption. Both metrics depend on the Inception-v3 Network [54], a pretrained ImageNet model, to calculate statistics on the generated output (for IS) and on the real and generated distributions (for FID). The validity of these metrics when applied to other datasets has been repeatedly called into question [3, 48, 8, 46]. Perturbations imperceptible to humans alter their values, similar to the behavior of adversarial examples [33]. Finally, similar to our metric, FID depends on a set of real examples and a set of generated examples to compute high-level differences between the distributions, and there is inherent variance to the metric depending on the number of images and which images were chosen—in fact, there exists a correlation between accuracy and budget (cost of computation) in improving FID scores, because spending a longer time and thus higher cost on compute will yield better FID scores [38]. Nevertheless, this cost is still lower than paid human annotators per image.

**Human evaluations.** Many human-based evaluations have been attempted to varying degrees of success in prior work, either to evaluate models directly [14, 42] or to motivate using automated metrics [52, 23]. Prior work also used people to evaluate GAN outputs on CIFAR-10 and MNIST and even provided immediate feedback after every judgment [52]. They found that generated MNIST samples have saturated human performance — i.e. people cannot distinguish generated numbers from real MNIST numbers, while still finding $21.3\%$ error rate on CIFAR-10 with the same model [52]. This suggests that different datasets will have different levels of complexity for crossing realistic or hyper-realistic thresholds. The closest recent work to ours compares models using a tournament of discriminators [42]. Nevertheless, this comparison was not yet rigorously evaluated on humans nor were human discriminators presented experimentally. The framework we present would enable such a tournament evaluation to be performed reliably and easily.

# 7    Discussion and conclusion

**Envisioned Use.** We created HYPE as a turnkey solution for human evaluation of generative models. Researchers can upload their model, receive a score, and compare progress via our online deployment. During periods of high usage, such as competitions, a retainer model [4] enables evaluation using $\text{HYPE}_\infty$ in 10 minutes, instead of the default 30 minutes.

**Limitations.** Extensions of HYPE may require different task designs. In the case of text generation (translation, caption generation), $\text{HYPE}_{\text{time}}$ will require much longer and much higher range adjustments to the perceptual time thresholds [29, 59]. In addition to measuring realism, other metrics like diversity, overfitting, entanglement, training stability, and computational and sample efficiency are additional benchmarks that can be incorporated but are outside the scope of this paper. Some may be better suited to a fully automated evaluation [8, 38]. Similar to related work in evaluating text generation [21], we suggest that diversity can be incorporated using the automated recall score measures diversity independently from precision $F_{1/8}$ [51].

**Conclusion.** HYPE provides two human evaluation benchmarks for generative models that (1) are **grounded** in psychophysics, (2) provide task designs that produce **reliable** results, (3) **separate** model performance, (4) are cost and time **efficient**. We introduce two benchmarks: $\text{HYPE}_{\text{time}}$, which uses time perceptual thresholds, and $\text{HYPE}_\infty$, which reports the error rate sans time constraints. We demonstrate the efficacy of our approach on image generation across six models {StyleGAN, SN-GAN, BigGAN, ProGAN, BEGAN, WGAN-GP}, four image datasets {CelebA-64, FFHQ-1024, CIFAR-10, ImageNet-5}, and two types of sampling methods {with, without the truncation trick}.

# Acknowledgements

We thank Kamyar Azizzadenesheli, Tatsu Hashimoto, and Maneesh Agrawala for insightful conversations and support. We also thank Durim Morina and Gabby Wright for their contributions to the HYPE system and website. M.L.G. was supported by a Junglee Corporation Stanford Graduate Fellowship. This work was supported in part by an Alfred P. Sloan fellowship. Toyota Research Institute ("TRI") provided funds to assist the authors with their research but this article solely reflects the opinions and conclusions of its authors and not TRI or any other Toyota entity.

## Footnotes

[2]We explicitly reveal this ratio to evaluators. Amazon Mechanical Turk forums would enable evaluators to discuss and learn about this distribution over time, thus altering how different evaluators would approach the task. By making this ratio explicit, evaluators would have the same prior entering the task.

[3]Hyper-realism is relative to the real dataset on which a model is trained. Some datasets already look less realistic because of lower resolution and/or lower diversity of images.

[4]We do not pursue time exposures under 100ms due to constraints on JavaScript browser rendering times.

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
