[Reviews · NeurIPS 2019]

Reviewer 1



This paper proposes a human based benchmark system in order to rate GANs. While readers might argue this is too costly, the authors nicely illustrates the costs of their system and motivate the need for using human raters as the true gold standard. While the system is introduced as being general and applicable to any type generative model, the author missed a (very recent) related work from the NLP community. The system HUSE from the paper Unifying Human and Statistical Evaluation for Natural Language Generation, Hashimoto et. al. 2019 utilizes a similar turing test like method to rate the accuracy of a translation model. In addition to this fact, the Hashimoto et. al. claim that, while humans are the gold standard for rating the accuracy, they miss to quantify the diversity capacity of an underlying generative model.

Reviewer 2



This paper introduces a framework to evaluate the perceptual realism of samples from generative models. The framework, HYPE- Human Eye Perceptual Evaluation, is based on psychophysics methods. Two different metrics are proposed. The first one, HYPE_time, measures the amount of time a human needs before distinguishing a real from a fake. The metric is clearly defined and very well founded on psychophysics. The second one, HYPE_infinite, measures the error rate of a human when wrongly classifying fake images and real images (given unlimited time). This second metric is much simpler, faster and cheaper (in terms of human labor) while maintaining the reliability of the first one. The paper is very well written, it is based on psychophysical theory, the methodology is meticulously detailed and the experimental results and conclusions are quite interesting. The work will clearly contribute to the research and development of better generative models, an issue of major importance in machine learning. I am undoubtedly in favor of acceptance. Nevertheless, in what follows I list a few comments I have: -- The two proposed metrics allow ranking models according to realism, but despite this, their absolute value or even their relative difference does not mean anything. The same amount of numerical change in HYPEx value does not correspond to the same amount of visually perceived change in realism. This implies that the proposed metrics do not inform how much better a model is, but only produce a ranking. I would like the authors to comment on this. -- The proposed metrics only measure sample realism. But another very important property of generative models is diversity. This is openly stated as a limitation of the method. Notwithstanding, I think it would be good to discuss how a measure of diversity could be incorporated into the framework (either from human evaluation or from automatic measurements of the generated samples). A ranking of generative models should contemplate both realism and diversity at least. -- Comparison to automatic metrics. In the end, evaluating a single generative model is rather costly so the only way out seems to be to compute automatic measurements. The authors compare to FID, KID, and F1/8 but there is not too much discussion regarding this. It would be interesting to improve this section by trying to draw conclusions a little more interesting beyond saying whether or not the metrics are correlated. Also, the section would look better if a figure with all the FID/HYPE/XX data points was shown not only the correlation coefficients. -- Regarding reproducibility. It would be good if the authors made available all the necessary data to recalculate the metrics shown in the paper. In particular, the evaluations of each human on the generated images that are used. This would allow other researchers to fully reproduce the results, and also facilitate to continue the research in different lines (e.g., can the humans be clustered in terms of realism perception? - do all humans behave more or less the same? Are some better correlated with any of the automatic metrics?) Other minor comments: -- During the evaluation procedure, when presenting images to the experts, in many cases, you mention that half of the images are fake and half are real (e.g., line 135 - 50real / 50fake). Knowing this proportion would bias the expert. Could this be a problem? -- Hyper-realism is hard to understand (the generator produces images that look more real than real ones). Maybe this is associated with poor quality image datasets (e.g., CIFAR 10) where the images might look a little artificial. Could you add a short comment on this? -- Table 1 (pag 5). There seems to be a typo in the first and second model since the HYPEtime values are out of the respective 95% CIs. ------ After rebuttal. I appreciate the answers and comments of the authors. I think this is a very interesting work that clearly deserves to be published in this venue.

Reviewer 3



In this work, two new benchmarks are introduced for better evaluation of generative models. The first metric is HYPE_time, which computes the minimum amount of time it takes a person to distinguish an image as real or fake. The second metric, HYPE_infinity, measures the errors of people given unlimited time, and is much faster to compute and more cost-effective. The authors test these two metrics on multiple datasets using many models and show that HYPE is reproducible and can be used to separate the efficacy of different models. Further, HYPE was shown to be a more reliable predictor than alternative automated measures. This metric can be used to more consistently compare different generative approaches and provide a foundation for research in this area. Strengths: - A benchmark for generative models can be very useful for the community to enable more consistent evaluation of new methods. - The authors provided a thorough set of experiments for the two metrics, HYPE_time and HYPE_infinity, across different models and datasets. Weaknesses: - There was no extended discussion on related work, even though there are other metrics that have been proposed for evaluating generative models. The authors should justify why their metrics are more effective and how they differ from other benchmarks. Originality: There are a few related works on developing benchmarks for generative models, as included below. It would be great to get a justification for how the proposed metrics are preferred. Xu, Qiantong, et al. "An empirical study on evaluation metrics of generative adversarial networks." arXiv preprint arXiv:1806.07755 (2018). Wang, Zhengwei, et al. "Neuroscore: A Brain-inspired Evaluation Metric for Generative Adversarial Networks." arXiv preprint arXiv:1905.04243 (2019). Quality: The quality of the work is high. The measures were based on prior research, which motivates the choice of HYPE. The authors tested a variety of models with many datasets and showed the consistency in the metrics’ prediction of model performance. Clarity: The paper was well-written. All of the included details about the datasets and models were helpful in understanding the evaluation of HYPE. There were a couple of typos: - Pg 5: the results of HYPE_infinity approximates the those from → the results of HYPE_infinity approximate those from - Pg 7: one for each of object classes → one for each of the object classes Significance: The addition of a benchmark for evaluating generative models can be very useful to the field, as having standardized metrics allow for more consistent comparison of models. Having reproducible measures are incredibly important for advancing research in this space. ----------------------- I read the author response and am satisfied with the authors' discussion about how the work differs from prior literature.

[Author Response · NeurIPS 2019]

We thank the reviewers for their enthusiastic feedback and insightful suggestions. We appreciate the recognitions of
contribution in the reviews, such as "the methodology is meticulously detailed, and the work's contributions to research
and development of better generative models, an issue of major importance in machine learning," that HYPE as a
"benchmark for generative models can be very useful for the community to enable more consistent evaluation of new
methods," and that there was "[s]olid writing and justification for HYPE as a benchmark." We also appreciate the
questions and concerns raised in the reviews, as well as the requests and opportunity for clarification in our author
feedback response. We address them as follows and in our corresponding revision.

**[R1, R2] Proposed diversity incorporation.** R1 and R2 ask how diversity may be incorporated into a full ranking of
the GANs. As mentioned in our submission and acknowledged by R1 and R2, the current scope of HYPE is limited to
perceptual realism, though we agree that diversity measurement is worthwhile. Similar to HUSE (Hashimoto 2019),
we suggest that diversity can be computed using the automated recall score (Sajjadi 2018). Recall measures diversity
independently from precision ($F_{1/8}$), the corresponding measure of fidelity in the paper. We will include this comment
in our paper, citing both works.

**[R3] Related Work.** We will update our draft with an extended discussion on related work, stating how HYPE differs,
beyond FID, KID, and $F_{1/8}$. Broadly, HYPE measures human perceptual judgments of generative outputs using humans
directly in an inexpensive, widely accessible method. This contrasts with Neuroscore (Wang 2019) because it can be
widely accessible by researchers rather than depend on humans with EEGs properly worn and measured. Additionally,
the automated metrics 1-NN and Kernel MMD, suggested for evaluation by Xu et al., show success only in a limited
architectural case (the convolutional space of an ImageNet-pretrained ResNet) and nevertheless indirectly assess human
judgment.

Specifically, we address limitations of automatic metrics. Prior work has asserted that there exists coarse correlation of
human judgment to FID and IS, leading to their widespread adoption. However, both metrics depend on an ImageNet-
pretrained Inception v3 Network to calculate statistics on the generated output (for IS) and on the real and generated
distributions (for FID). The validity of these metrics when applied to other datasets has been repeatedly called into
question (Barratt 2018, Rosca 2017, Borji 2018, Ravuri 2018). Perturbations imperceptible to humans alter their values,
similar to the behavior of adversarial examples (Kurakin 2016). Finally, because FID is a biased estimator, there is
inherent variance to the metric depending on the number of images and which images were chosen—in fact, there exists
a correlation between accuracy and budget (cost of computation) in improving FID scores, because spending a longer
time and thus higher cost on compute will yield better FID scores (Lucic 2018). KID addresses this as an unbiased
estimator (Bińkowski 2018), but otherwise strongly correlates with FID and not human judgment, as we report in the
paper.

**[R1] Impact of number of samples per evaluator.** Similar to our earlier evaluation on increasing the number of
evaluators, we find that the CI decreases when increasing the number of samples that each evaluator assesses. Computed
via a bootstrap, find that the CI width decreases monotonically when increasing the number of images, specifically
from $10.5$ to $8.5$ when evaluating $20$ versus $100$ images. We choose $100$ images for HYPE infinity because we find it
to be a good combination of reliability and efficiency for the GANs we evaluated. This number can be treated as a
hyperparameter, and may be increased or decreased based on the desired reliability and quality of GANs evaluated.

**[R2] Meaning of HYPE scores.** As R2 points out, HYPE is designed to rank GANs; deltas in HYPE scores may not
reflect an equal delta in visually perceived realism. However, we note that HYPE scores are directly correlated with
perceived realism, though that correlation may not be linear. Additionally, an absolute HYPE infinity score above $50$
meaningfully indicates "hyper-realistic" images, or images that appear more realistic than real images.

**[R2] Discussion on differences from automated metrics.** R2 requests an extended discussion on results from
automated metric differences. We will include comments in the paper on qualitative differences that we believe exist
between HYPE scores and {FID, KID, $F_{1/8}$} scores.

**[R2] Dataset and code release.** We plan to release the dataset and code for researchers to analyze after publication.

**[R2] Explicit 50:50 real:fake ratio.** R2 mentions that revealing the ratio to evaluators may bias them. We made this
design decision intentionally and carefully. Amazon Mechanical Turk forums would enable evaluators to discuss and
learn about this distribution over time, thus altering how different evaluators would approach the task (this discussion
occurs often). Thus, we decided to make this ratio explicit such that evaluators would have the same prior entering the
task. We also make this decision to evaluate a threshold for which fake images appear more realistic than the real ones.

**[R2] Hyper-realism with unrealistic real datasets.** R2 mentions the unrealistic appearance of some real datasets such
as CIFAR-10. We acknowledge this, and it enables varying levels of difficulty across datasets. Thus, "hyper-realism"
is relative to the real dataset on which a model is trained. Some are easier because of lower resolution and/or lower
diversity of images.

[Meta-Review · NeurIPS 2019]

The reviewers were unanimous in judging that this is good quality work that tackles a an important and relevant problem for NeurIPS, and that it will attract attention of a wide audience. The rebuttal solidified this viewpoint in the discussions thereafter. Given the enthusiastic reviews, I think this deserves an oral presentation at NeurIPS. [This meta-review was reviewed and revised by the Program Chairs]